# Colonic Volume Changes in Paediatric Constipation Compared to Normal Values Measured Using MRI

**DOI:** 10.3390/diagnostics11060974

**Published:** 2021-05-28

**Authors:** Hayfa Sharif, Caroline L. Hoad, Nichola Abrehart, Penny A. Gowland, Robin C. Spiller, Sian Kirkham, Sabarinathan Loganathan, Michalis Papadopoulos, Marc A. Benninga, David Devadason, Luca Marciani

**Affiliations:** 1Translational Medical Sciences, NIHR Nottingham Biomedical Research Centre at Nottingham University Hospitals NHS Trust, University of Nottingham, Nottingham NG7 2UH, UK; Hayfa.sharif@nottingham.ac.uk (H.S.); nichola.abrehart@nottingham.ac.uk (N.A.); robin.spiller@nottingham.ac.uk (R.C.S.); 2Ministry of Health, Civil Service Commission, Amiri Hospital, Kuwait City 15300, Kuwait; 3Sir Peter Mansfield Imaging Centre, School of Physics and Astronomy, University of Nottingham, Nottingham NG7 2RD, UK; caroline.l.hoad@nottingham.ac.uk (C.L.H.); penny.gowland@nottingham.ac.uk (P.A.G.); 4Nottingham Children’s Hospital, Nottingham University Hospitals NHS Trust, Nottingham NG7 2UH, UK; sian.kirkham@nuh.nhs.uk (S.K.); sabarinathan.loganathan@nuh.nhs.uk (S.L.); Michalis.Papadopoulos@nuh.nhs.uk (M.P.); david.devadason@nuh.nhs.uk (D.D.); 5Department of Pediatric Gastroenterology, Emma Children’s Hospital, Amsterdam UMC, 9, 1105 AZ Amsterdam, The Netherlands; m.a.benninga@amsterdamumc.nl

**Keywords:** colon, children, constipation, MRI, volume, transit

## Abstract

Background: Functional constipation in children is common. Management of this condition can be challenging and is often based on symptom reports. Increased, objective knowledge of colonic volume changes in constipation compared to health could provide additional information. However, very little data on paediatric colonic volume is available except from methods that are invasive or require unphysiological colonic preparations. Objectives: (1) To measure volumes of the undisturbed colon in children with functional constipation (FC) using magnetic resonance imaging (MRI) and provide initial normal range values for healthy controls, and (2) to investigate possible correlation of colonic volume with whole gut transit time (WGTT). Methods: Total and regional (ascending, transverse, descending, sigmoid, and rectum) colon volumes were measured from MRI images of 35 participants aged 7–18 years (16 with FC and 19 healthy controls), and corrected for body surface area. Linear regression was used to explore the relationship between total colon volume and WGTT. Results: Total colonic volume was significantly higher, with a median (interquartile range) of 309 mL (243–384 mL) for the FC group than for the healthy controls of 227 mL (180–263 mL). The largest increase between patients and controls was in the sigmoid colon–rectum region. In a linear regression model, there was a positive significant correlation between total colonic volume and WGTT (*R* = 0.56, *p* = 0.0005). Conclusions: This initial study shows increased volumes of the colon in children with FC, in a physiological state, without use of any bowel preparation. Increased knowledge of colonic morphology may improve understanding of FC in this age group and help to direct treatment.

## 1. Introduction

Functional constipation (FC) in childhood is common, with a prevalence estimated around 10% [1]. Constipation can substantially impair quality of life, with marked effects on the child’s social functioning [2]. It can result in repeated consultations with specialist consultants, unsuccessful treatment attempts, and the child enduring various investigations. Despite its prevalence, the aetiology of paediatric constipation is still poorly understood [3,4], and diagnosis principally is based on symptom reports alone [5,6]. Increased knowledge of colon physiology and morphology could provide healthcare professionals with additional, objective parameters to evaluate.

The length of the colon increases with age from around 1 m in children aged 4–6 years old [7] to approximately 1.5 m in adults. The colon is able to accommodate varying volumes of intraluminal contents [8]. It performs the last stage of digestion in the gastrointestinal system, including autonomic motor function, transport and fluid absorption, and storage of waste until defecation. Despite our understanding of the importance of the colon in gastrointestinal pathophysiological processes, relatively little is known about colonic volume in children with FC.

In previous reports, the volume of the adult human colon, either as a whole or divided in anatomical segments, has been estimated using different measurement techniques such as cadaver dissection studies [9,10], perioperative assessment [11], abdominal computed tomography (CT), or barium contrast studies [12,13]. However, these techniques are invasive, often require preparation of the colon and use contrast media, which can disturb the physiology, or use ionising radiation, which is particularly undesirable in children.

Recently, studies of the volume of the adult colon have been carried out using magnetic resonance imaging (MRI). The advantages of MRI are excellent soft tissue contrast, cross sectional image quality, speed, and the lack of use of ionising radiation, which can provide comprehensive 3D views of each colonic segment (ascending, transverse, descending colon, and sigmoid–rectum) without preparation or use of contrast agents for the measurement of colonic volume [14,15,16]. Several studies in adults have demonstrated the responsiveness of the total colonic volume measured by MRI, which, in randomised placebo controlled trials, showed the expected increase in response to bulking agents like psyllium and kiwifruit [17,18]. Intraluminal volumes have been measured before and after feeding interventions [16,18] and treatment with bulking agents and laxatives [17,19] to assess the colon volume functional responses. Moreover, MRI can study the colon in a physiological state without the need of bowel preparations to cleanse or distend the organ. Studies in these physiological conditions have reported increased volume of the colon in adults with FC and irritable bowel syndrome (IBS) compared to healthy controls [20,21]. They also showed that colonic volume in healthy subjects decreased by around 1/3 after defecation, indicating that delayed defecation would be predicted to significantly increase total colonic volume [22,23]. Delayed defecation with chronic faecal impaction is a common feature of FC in children, but to what extent it would increase colonic volume and correlate with transit has until now been unclear.

Therefore, the primary aim of this study was to measure colonic volume in children with FC and also provide initial normal range values for healthy controls. We used MRI images from a recently published study that measured whole gut transit time (WGTT) [24], which also allowed us to consider possible correlation of the new colonic volume data with WGTT as the secondary aim of this study.

## 2. Materials and Methods

### 2.1. Subjects and Study Design

The data for this retrospective study are from a previous MRI study investigating WGTT [24]. Participants had to be between the ages of 7 and 18 years old. The Rome IV criteria [5,6] were used to identify patients with paediatric FC following a referral either from primary or secondary care into a specialist clinic. Exclusion criteria included gastrointestinal surgery that could affect function, antegrade colonic enema (ACE) patients, inability to lie still for less than 5 mins, and presence of metallic implants or metallic foreign body in the eyes. Healthy controls were recruited from the local population using advertisements. The participants ingested 24 mini-capsule MRI markers each day for 3 days, and were imaged on day 4 and day 7 (a common radiopaque marker protocol) using MRI. The data used here were from the first MRI scan at day 4. The study was a methods feasibility study and as such, no treatment or lifestyle changes were requested of the participants. Similarly, no questionnaires about defecation were collected. The study was approved by the UK National Research Ethics Committee (17/WM/0049) and the UK Medicines and Healthcare Products Regulatory Agency (MHRA) (CI/2017/0054), and was registered on Clinicaltrials.gov (accessed on 1 January 2021) (NCT03564249). All participants below the age of 16 gave written assent to the study and their parent/carer gave informed written consent of participation. All participants between 16 and 18 years old gave informed written consent to the study. The procedure included permission to use the data for further research.

### 2.2. Magnetic Resonance Imaging

All MRI scans were carried out using 3T MRI scanners (Philips, Best, The Netherlands), sited at the Sir Peter Mansfield Imaging Centre at the University of Nottingham. The subjects were not sedated and lay feet first on the scanner bed, in the supine position with a 16-channel abdominal receiver coil wrapped around the abdomen. A short breath-hold multiple-echo (mDIXON) sequence [25] (echo times 1/2 = 1.32/2.2 ms, repetition time = 10 ms, flip angle = 20°, field of view = 250 × 350 mm^2^, acquired resolution = 1.8 × 1.8 × 4.4 mm^3^) was used to acquire both axial and coronal views of the abdomen, divided into short breath-hold (approximately 13 s) stacks. The whole MRI procedure took approximately 15 to 20 min. The MRI scans were retrieved and analysed as detailed below.

### 2.3. Data Analysis and Statistics

The coronal mDIXON fat and water in-phase images were used to measure the colonic volume, in keeping with previous work in adults [16,21,26]. Individual regional colon volumes were manually segmented on each slice using Medical Image Processing, Analysis, and Visualisation software (MIPAV, NIH, Bethesda). The data analysis was not blinded. The colon of the participants was divided into 4 regions: ascending colon (AC), transverse colon (TC), descending colon (DC), and sigmoid colon and rectum region (SC-R). Regional boundaries commenced at the caecum (ascending) and were fixed in a coronal plane at the superior point of the hepatic flexure (ascending to transverse), at the splenic flexure (transverse to descending) and terminated at the sagittal plane of the commencement of sigmoid colon where the descending colon deviated posteriorly or medially. Each colon region was identified within each coronal image slice, and a region of interest (ROI) drawn around it, building a 3D representation of the morphology from which the volume of each region was measured by summing up the corresponding ROIs and multiplying by slice thickness. If any difficulties were found in visualising the anatomy, axial views were also used to aid navigation. The colon volume data were corrected for body surface area (BSA) using the Mosteller formula [26,27]. The formula calculates BSA as the square root of (height (cm) × weight (kg)/3600). Linear regression was used to explore the relationship between total colon volume from this study and the individual participants’ WGTT values from [24].

Statistical analysis was carried out using Prism 8 (GraphPad Software Inc., La Jolla, CA, USA). Normality of the data was checked using the Shapiro–Wilk normality test. Depending on normality, comparisons between groups were performed using two-tailed *t*-test or Mann–Whitney rank sum test for unpaired data. Differences were considered significant at *p* < 0.05.

## 3. Results

Nineteen healthy volunteers with no history of gastrointestinal disease (8 male; 11 female; age 16 ± 2 years old; body mass index (BMI) 25 ± 5 kg/m^2^) and 16 patients with functional constipation (7 male; 9 female; age 11 ± 3 years old; BMI 25 ± 9 kg/m^2^) participated in the study. The difference in age between groups was statistically significant (Mann–Whitney test, *p* < 0.0001). Good-quality images of the colon were obtained in the majority of the 35 participants. The images from 4 participants displayed some breathing artefacts. This, however, did not prevent identification and mapping of the colon. All data from the 35 participants were included in the image analysis. There was a moderate positive linear correlation of colon volume with height (*R* = 0.49 for patients, *p* = 0.058; and *R* = 0.60 for controls, *p* = 0.0068) and weight (*R* = 0.45 for patients, *p* = 0.0802; and *R* = 0.30 for controls, *p* = 0.2163). Moreover, the height and weight of the young participants varied markedly, with some being nearly twice as tall (range 105 cm to 185 cm) and/or 4 times heavier (range 22 kg to 106 kg) than others, which underpinned the use of the colonic volume correction for body surface area.

Example colon reconstructions are shown in Figure 1. A 13-year-old patient with FC with a BMI of 17.7 kg/m^2^ had the largest total colon volume in the patient group (712 mL), with a WGTT of 140 h. This patient presented with a megarectum (Figure 1a), with a sigmoid colon and rectum volume of 243 mL. By comparison, Figure 1b shows the colon reconstruction for a 17-year old-healthy control with a BMI of 23.4 kg/m^2^ who had the largest colon volume (360 mL) among the control group and a WGTT of only 16 h. This control participant was one of the tallest ones, with a height of 185 cm.

The uncorrected colonic volume had some degree of correlation with participants’ height (Spearman’s r = 0.31, *p* < 0.1) and with participants’ weight (between colonic volume and weight (Spearman’s r = 0.38, *p* < 0.05).

The total colonic volume corrected for BSA (Figure 2) was 309 mL (243–384 mL) median (interquartile range) for the young patients with constipation, significantly larger than that for the healthy controls’ 227 mL (180–263 mL) (*p* = 0.0081 Mann–Whitney two-tailed test). The data showed that 63% of patients had a colonic volume above the 75% centile of normal control values, and 25% of patients had the colon volume above the 95th centile (upper limit of normal) of the healthy control values.

The regional colon volumes presented in Figure 3 showed that the larger increase between controls and patients was in the SCR-region (*p* = 0.0410, Mann–Whitney one-tailed test), with 9 constipated patients having SCR-volume above the 95% CI of the control values.

Median and IQR data of the colon of controls and patients are summarised numerically in Table 1.

There was a positive correlation between the total colonic volume and WGTT (Figure 4); coefficient of correlation *R* = 0.56, *p* = 0.0005.

## 4. Discussion

Colon volume has been shown to be increased in adult functional constipation (FC) [20,21]. This is the first comparative study on the undisturbed colonic volume in paediatric patients with FC and healthy paediatric controls using MRI. An early paper from 1973 investigated colonic volume in idiopathic constipation, but the volume was derived from invasive constant infusion studies on unphysiologically lavaged colons [28]. In that report, adult data were mixed with the data from the paediatric participants, making it difficult to compare.

In our study, the total colon volume was found to be larger in the paediatric patients with FC than in healthy controls. As expected, the data also showed some degree of correlation between colonic volume and height and between colonic volume and weight. These considerations led to the use of a common Mosteller correction for body surface area [27] to allow for a more realistic comparison between organ volumes, as previously reported for MRI colon volume in children and adults with cystic fibrosis [26].

We also found a moderate but significant correlation of transit with colonic volume. Intervention studies in healthy adults and adults with constipation showed that feeding bulking agents such as ispaghula [17] and kiwifruit [18] increased colonic volume, most obviously in the proximal colon, but these agents did not alter transit. Defecation has been shown to reduce colonic volume by around 1/3, mostly after emptying the distal colon [22]. One can infer that defecation would influence the total colon volume of children with FC more, since the distal colon represents a bigger proportion of their total colon volume than in adults. For most adult subjects, colonic volume is determined by many factors other than whole gut transit time. In a recent, prospective MRI study in 20 adult patients with constipation, the maximum luminal colon diameter was measured from the images in different colonic segments [29]. The measurements were then correlated with gastrointestinal symptom rating scales. Interestingly, the authors reported several correlations of colonic diameters with symptom scores, including between the diameter of the descending colon and unsatisfactory defecation, and between the rectum and constipation scores. In our study, we did not collate symptom questionnaires on the MRI study day, and therefore any correlation between paediatric colon volumes and symptoms in our paediatric cohort remains to be investigated. Constipation can also be associated with changes in microbiota [30], so metabolic changes and influence on fermentation cannot be excluded. Limitations of this study included the post hoc analysis of images from a previous study [24], carried out unblinded and with a lack of correlation with clinical data, which the original feasibility study did not collect. This may increase the likelihood of chance findings, although the differences shown here had a high degree of statistical significance. Another limitation of this study was that, while the age ranges of the two groups were similar, the healthy participants were older.

These novel insights are also relevant for the pharmaceutical industry. Targeted drug-delivery systems are being investigated for their potential in treating colonic diseases directly through an intraluminal approach [31]. Increased knowledge of paediatric colonic volume will allow pharmacokinetic modelling parameters to be estimated more accurately, making drugs more bespoke to the recipient.

## 5. Conclusions

This study showed increased colon volumes in children with functional constipation compared to healthy controls. The volume increase in the patients was particularly marked in the sigmoid–rectum region. MRI allowed us to investigate colon volume in the paediatric population in a physiological state, without using any bowel preparation. Further work is needed to increase the number of participants studied and continue to investigate the relationship between colonic volume and WGTT. Increased knowledge of colonic abnormalities may improve understanding of this condition and help to direct treatment.

## Figures and Tables

**Figure 1 diagnostics-11-00974-f001:**
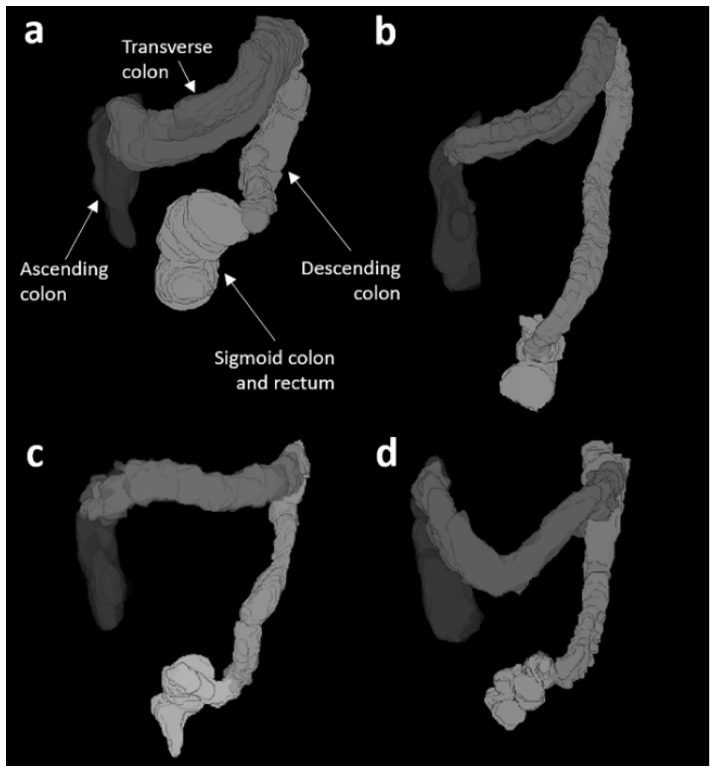
3D representations of the colon morphology from four different participants whose characteristics were: (**a**) patient, 13 years old, weight 42 kg, height 154 cm, BMI 17.7 kg/m^2^, total colon volume (TCV) = 712 mL, whole gut transit time (WGTT) = 140 hs; (**b**) healthy control, 17 years old, weight 80 kg, height 185 cm, BMI = 23.4 kg/m^2^, TCV = 360 mL, WGTT = 16 hs; (**c**) patient, 9 years old, weight 48 kg, height 130 cm, BMI 21.4 kg/m^2^, TCV= 237 mL, WGTT = 75 hs; (**d**) healthy control, 10 years old, weight 42 kg, height 145 cm, BMI = 19.9 kg/m^2^, TCV = 227 mL, WGTT = 26 hs.

**Figure 2 diagnostics-11-00974-f002:**
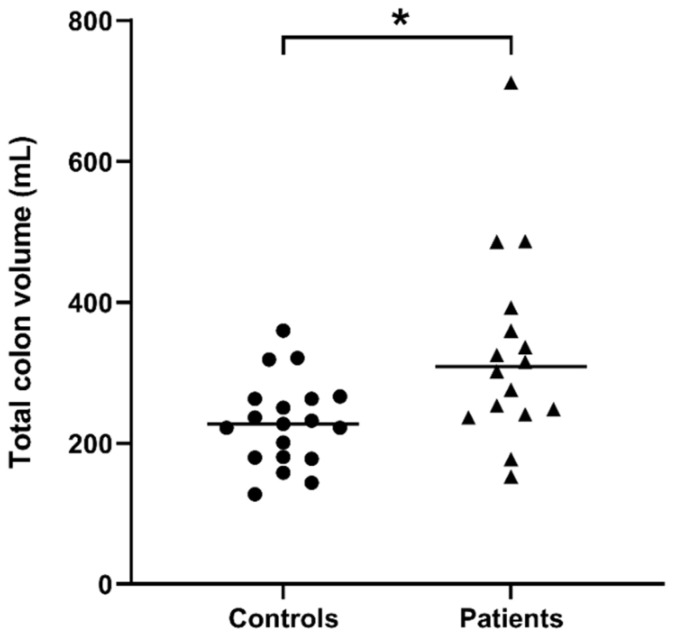
Total colon volume corrected for both body surface area of *n* = 19 controls and *n* = 16 patients with constipation. The horizontal lines indicate the median, * *p* = 0.0081 two-tailed Mann–Whitney test.

**Figure 3 diagnostics-11-00974-f003:**
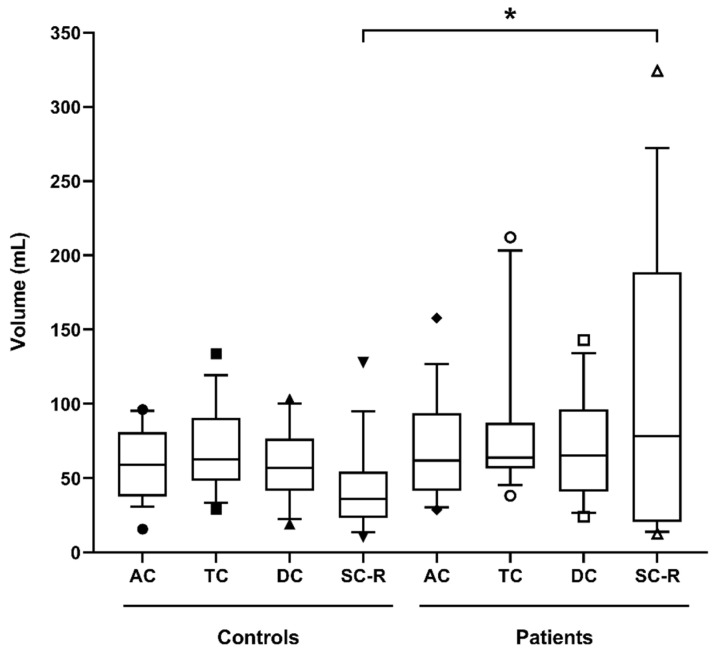
Regional colon volumes corrected for body surface area (ascending colon (AC), transverse colon (TC), descending colon (DC), sigmoid colon and rectum (SC-R)) for 19 controls and 16 patients with constipation. The box-and-whiskers plot shows the mean (+) and median (−) values of the regional ascending, transverse, descending, and sigmoid colon and rectum volumes. The box represents the 25th–75th centile, and the whiskers represent the 10th–90th centile ranges, respectively. * *p* = 0.0410 one-tailed Mann–Whitney test.

**Figure 4 diagnostics-11-00974-f004:**
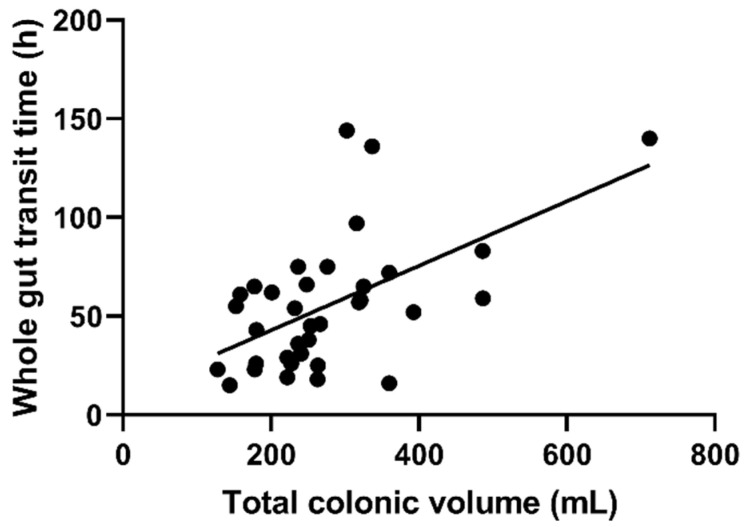
Correlation between whole gut transit time (WGTT) and total colonic volume, corrected for body surface area, for 19 controls and 16 patients with constipation coefficient of correlation *R* = 0.56, *p* = 0.0005.

**Table 1 diagnostics-11-00974-t001:** Regional and total colon volumes corrected for body surface area for ascending colon, transverse colon, descending colon, and sigmoid colon and rectum for *n* = 16 patients with functional constipation and *n* = 19 controls. The data are presented in median (interquartile range). *p*-values versus control are indicated in the last line of the table; n/s = not significant.

	Ascending Colon (mL)	Transverse Colon (mL)	Descending Colon (mL)	Sigmoid Colon and Rectum (mL)	Total Colon (mL)
Controls	59 (38–81)	63 (48–91)	57 (41–77)	36 (23–54)	227 (180–263)
Patients	62 (41–94)	64 (56–87)	65 (41–96)	78 (20–189)	309 (243–384)
*p*-value	n/s	n/s	n/s	*p* < 0.05	*p* < 0.001

## Data Availability

The data presented in this study are available on request from the corresponding author. The data are not publicly available yet due to ongoing research work.

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
