# Peer review of "Colonic Volume Changes in Paediatric Constipation Compared to Normal Values Measured Using MRI"

_diagnostics, 2021, doi:10.3390/diagnostics11060974_

Round 1

Reviewer 1 Report

The manuscript entitled "Colonic volume changes in paediatric constipation compared to normal values measured using MRI" was reviewed.  The number of samples is small, but the research content is interesting.

Please clearly indicate the exclusion criteria.
Please describe in detail when to take an MRI, especially about defecation, eating and taking medicine. 

There is a great deal of knowledge about MRI of chronic constipation in adults. Please consider the difference between adults and children in more detail by referring to the following paper.

Inoh Y, et al. Assessment of colonic contents in patients with chronic constipation using MRI. J Clin Biochem Nutr. 2018 May;62(3):277-280. doi: 10.3164/jcbn.17-104. 

Author Response

We are grateful to this Reviewer for their comments. We have answered all the points raised as detailed point-by-point below. All changes were fully tracked in Word and are also identified below by the new line numbers as seen using the ‘All Markup’ reviewing option on the Word manuscript version R1.

The manuscript entitled "Colonic volume changes in paediatric constipation compared to normal values measured using MRI" was reviewed.  The number of samples is small, but the research content is interesting.

ANSWER: Thank you for your positive appraisal of our manuscript.

Please clearly indicate the exclusion criteria.

ANSWER: agreed, we have now added more details of inclusion and exclusion criteria at lines 96-102.

Please describe in detail when to take an MRI, especially about defecation, eating and taking medicine.

ANSWER: Thank you for raising this point, we have added an explanation (at lines 104-107) that the original study was a methods feasibility study and as such no treatment or lifestyle changes were requested of the participants, and no questionnaires about defecation timings were collected. We have also indicated the timing of the MRIs at Day 4 and Day 7, with the first scan used for this new work.

There is a great deal of knowledge about MRI of chronic constipation in adults. Please consider the difference between adults and children in more detail by referring to the following paper.

Inoh Y, et al. Assessment of colonic contents in patients with chronic constipation using MRI. J Clin Biochem Nutr. 2018 May;62(3):277-280. doi: 10.3164/jcbn.17-104.

ANSWER: Thank you for raising this point and for suggesting this paper. We had tried to keep the manuscript focused primarily on pediatric constipation but we agree this had limited the consideration of the difference against adults. We have now added the suggested reference and expanded a comment to this effect in the Discussion at lines 257-265. In our study, we did not collate symptom questionnaires on the MRI study day and therefore any correlation between pediatric colon volumes and symptoms in our pediatric cohort remains to be investigated.

Reviewer 2 Report

An interesting, however erratic, article. The major concerns are:

  1. Informed consents need clarification.
  2. References 5 and 14 should be removed (unnecessary self-citation).
  3. Methods, Results, and Discussion chapters need re-editing--there are results in Methods and Discussion. The Results are chaotic.
  4. Numerous flaws as in the attached file. Please, ensure the authors have an access to it.

The authors proved that in constipated children feces accumulate in the colon--but we know it, even without MRI.

The aims of the study should be changed.

Author Response

We are very grateful to this reviewer for the time taken to review our paper and for their comments, which we believe have improved our manuscript substantially. We have answered all the points raised as detailed point-by-point below. All changes were fully tracked in Word and are also identified below by the new line numbers as seen using the ‘All Markup’ reviewing option on the Word manuscript version R1.

An interesting, however erratic, article. The major concerns are:

ANSWER: Thank you so much for this thorough and constructive revision. We are grateful for the overall comment that our work is interesting and we appreciate all the suggestions for improvement given. We have taken all of those on board and we have answered all the points raised as detailed point-by-point below. All changes were tracked in Word and are identified below also by the new line number of the ‘tracked changes’ Word manuscript version R1.

Informed consents need clarification.

ANSWER: Agreed. We have now provided more details about the process of written assent below the age of 16 (plus written parental/carer consent) and consent between 16-18 years of age at lines 109-114.

References 5 and 14 should be removed (unnecessary self-citation).

ANSWER: Agreed. Reference 5 and 14 were both removed.

Methods, Results, and Discussion chapters need re-editing--there are results in Methods and Discussion.

ANSWER: thank you for the suggestions and the very useful PDF comments), we have followed them and moved all those parts back into Results as detailed more specifically below

The Results are chaotic.

ANSWER: Thanks to the Reviewer’s suggestions we have now moved back all results in the Results section and we hope that the rearrangement now answers this point too.

Numerous flaws as in the attached file. Please, ensure the authors have an access to it.

ANSWER: Thank you for the comments on the PDF file we have answered all the points in the PDF as detailed below.

The authors proved that in constipated children feces accumulate in the colon--but we know it, even without MRI.

ANSWER: We agree that feces accumulation/impaction and particularly sigmoid rectum retention are known, it also true that there are very little quantitative data in children with constipation. Little is known particularly about the extent to which this would increase colon volumes and whether there may be a relationship between colon volume and colon transit, this is now clarified more overtly at lines 81-84.

The aims of the study should be changed.

ANSWER: we hope the point above is clarified, it then follows that the aims of the study are justified. We have however tidied up the paragraph by removing the confusion about ‘different ages’.

SPECIFIC COMMENTS ON PDF

Line 32 Please delete

ANSWER: Agreed, the acronym IQR was deleted from line 33 and also deleted the p values in parentheses at lines 34 and 35.

Line 35 It is misleading. You used linear regression model, not Pearson or Spearman correlation. Perhaps it would be of value to add: "In a regression model, ..." Discuss it with your statistician.

ANSWER: Agreed we have added at line 35 ‘In a linear regression model ...’

Line 59-60 These are highlighted but there is no comment appended to the PDF

ANSWER: We interpreted the highlighting as part of the comment on the need to expand on the notion of ‘un-prepared’ colon, which we have now done

Line 74 - please, add explanation of these notions

ANSWER: we have added an explanation to this effect at lines 75-78 to clarify that the colon can be studied in a physiological state without the need of bowel preparations to cleanse or distend the or-gan. This, accordingly, has also been clarified in Abstract line 38.

Line 77 - They are symptoms of FC, not the cause. As you mentioned previously--we do not know all the causes.

ANSWER: agreed, we have substituted the term ‘important cause of FC’ with ‘common feature of FC’ at line 82.

Line 79 It is the opposite in the cited article. The authors even propose to define constipation as chronic feacal retention. What is not known is the size of the feacal retention and correlation with transit time (WGTT).

ANSWER: Agreed, we have removed the reference and clarified that ‘to which extent it would increase colonic volume and correlate with transit has until now been unclear’ at lines 82-84

Line 88 –

  1. These are results. Instead of numbers, please provide the readers with inclusion and exclusion criteria.

ANSWER: Agreed, we have now added more details of inclusion and exclusion criteria at lines 96-102.

  1. Figures should be moved to the results chapter.

ANSWER: agreed, the figures of number and demographics of the participants have been moved to the beginning of the Results section lines 154-157.

  1. Were there statistically significant differences between groups regarding age?

ANSWER: thank you for this comment, we have added at lines 157-158 that the difference in ages was statistically significant and also added a comment to this effect in the limitations in the Discussion at lines 270-272.

Line 90 - Are the Rome IV criteria validated for children

ANSWER: The paediatric Rome IV criteria are established criteria to diagnose children of all ages with functional gastrointestinal disorders including constipation. Whilst there are validated questionnaires to diagnose these disorders, they are not needed to make the diagnosis.

Line 94 - Was it children's consent or proxy? It must be clear.

ANSWER: This points echoes the one in the Reviewer’s web form text above, we have now provided more details about the process of written assent below the age of 16 (plus written parental/carer consent) and consent between 16-18 years of age at lines 109-114.

Line 125 and/or 132 – Duplication

ANSWER: Thank you for spotting this, we have removed the duplication at lines 151-152 (previous manuscript line 132).

Line 132 - specifically: in 31 out of 35 participants.

ANSWER:  this line has been removed as a duplication. In answer to the specific point we have n=35 data points for the correlation. We mention in Results that there was some breathing artefact in the images from 4 participants but also commented that this did not prevent us measuring the colon volumes for this work.

Line 188 - Please add the line with p values (n/s, p <0.05, p <0.001) for all volumes.

ANSWER: Thank you for the suggestion, the line with p values has been added to Table 1 at lines 221-222.

Line 210 - These are results and should be placed in the previous chapter.

The reader should not be startled by new figures not presented previously.

ANSWER: Thank you, we agree, the numerical results have been moved to the Results section lines 185-187 and in Discussion only the general comment has been left at lines 239-240.

Line 212 - This is repetition--you have already stated that this is the first such report, so the data need to be the only one.

ANSWER: Agreed, we have removed the sentence ‘There are no similar studies to compare our data against’ at lines 242-243.

Line 215 - You did not mention the stratification in the results.     

ANSWER: Thank you for pointing this out, it was not meant to say we sub-divided the participants in sub-groups of age. We have removed the ‘by age groups’ words from lines 245-246. We have also added the correlation values in the Results at lines 185-187.

Line 219 - I cannot aggree. Only for healthy children.

ANSWER: This point raised made us go back to the raw data and double-check what we had done. We would like to reassure the Reviewer that all n=35 point, both patients and healthy participants are included in Figure 4 and that the relationship does show a positive correlation between the total colonic volume and WGTT (Figure 4), with a coefficient of correlation R=0.56, p=0.0005 – we have also tried Spearman’s which provides a similar answer with Spearman’s r=0.46 and p=0.0055.

Line 233 - I think it is wrong.

It does not refer to colon. Bioavailability depends on small intestine.

ANSWER: We have removed the ‘bioavailability’ term and refer now to general ‘pharmacokinetic modelling parameters’ at line 276 – drug dissolution, transport and absorption will depend on chyme volumes and modelling will benefit from improved boundary conditions

Line 239 - Two-three key statement based on the relevant findings rather than repeating that it is a novel study.

ANSWER: Thank you for this comment, we have re-written the Conclusions as suggested and without repeating that it is a new study at lines 279-288.

Round 2

Reviewer 2 Report

Thank you for all amendments and proofs. Perfectly done!